# Mapping the Morbidity Risk Associated with Coal Mining in Queensland, Australia

**DOI:** 10.3390/ijerph19031206

**Published:** 2022-01-21

**Authors:** Javier Cortes-Ramirez, Darren Wraith, Peter D. Sly, Paul Jagals

**Affiliations:** 1School of Public Health and Social Work, Queensland University of Technology, Brisbane, QLD 4059, Australia; d.wraith@qut.edu.au; 2Children’s Health and Environment Program, Child Health Research Centre, The University of Queensland, Brisbane, QLD 4101, Australia; p.sly@uq.edu.au (P.D.S.); p.jagals@uq.edu.au (P.J.)

**Keywords:** coal mining, morbidity, Integrated Nested Laplace Approximation (INLA), spatial regression, area specific risk, cardiovascular diseases, respiratory diseases

## Abstract

The populations in the vicinity of surface coal mining activities have a higher risk of morbidity due to diseases, such as cardiovascular, respiratory and hypertensive diseases, as well as cancer and diabetes mellitus. Despite the large and historical volume of coal production in Queensland, the main Australian coal mining state, there is little research on the association of coal mining exposures with morbidity in non-occupational populations in this region. This study explored the association of coal production (Gross Raw Output—GRO) with hospitalisations due to six disease groups in Queensland using a Bayesian spatial hierarchical analysis and considering the spatial distribution of the Local Government Areas (LGAs). There is a positive association of GRO with hospitalisations due to circulatory diseases (1.022, 99% CI: 1.002–1.043) and respiratory diseases (1.031, 95% CI: 1.001–1.062) for the whole of Queensland. A higher risk of circulatory, respiratory and chronic lower respiratory diseases is found in LGAs in northwest and central Queensland; and a higher risk of hypertensive diseases, diabetes mellitus and lung cancer is found in LGAs in north, west, and north and southeast Queensland, respectively. These findings can be used to support public health strategies to protect communities at risk. Further research is needed to identify the causal links between coal mining and morbidity in non-occupational populations in Queensland.

## 1. Introduction

The global demand for energy has determined the growing production and use of coal in the last 60 years with increasing potential for negative health impacts on people in the vicinity of expanding coal mining activities. Coal mining emissions are associated with health effects in multiple body organ systems due to exposure to coal dust and direct and indirect responses to fine particles [1]. The impacts of particles released from coal mining on respiratory health have been known for more than a century, especially in coal miners who have an increased risk of chronic respiratory diseases, lung cancer and coal mine dust lung disease [2,3]. Coal dust that enters the lungs of occupationally exposed populations induces an inflammatory response determined by the aggregation of carbon-laden macrophages and can form nodular lesions within the lung [4]. Coal mining activities release particulate matter (e.g., PM_10_ and PM_2.5_) that can be associated with diseases of the circulatory and respiratory systems [5,6]; these particles include metals that are classified as carcinogens [7,8]. Other releases from coal mining include fine particle-bounded polycyclic aromatic hydrocarbons (PAHs) that have been associated with adverse effects in respiratory organs [9,10]. The mix of particles released in coal mining activities can determine diseases in exposed populations because of their ability to reach the smallest sections of the respiratory system and their potential effect on the endothelium and the inflammatory response [11,12].

Increasing research in the last two decades has found that non-occupational populations in the vicinity of surface coal mining activities are also at higher risk of increased mortality and morbidity [13,14]. A higher mortality due to cancer and cardiovascular, kidney, and respiratory diseases has been found in populations in counties with active open surface coal mines in the United States-USA [15,16,17]. These populations have increased morbidity associated with coal mining, including higher incidence of cancers of the colon [18], lung, and breast [3,19], and respiratory diseases [20], diabetes mellitus [21] and hypertensive diseases [22]. Children in coal mining areas in the United Kingdom have an increased risk of respiratory diseases and other conditions, such as diseases of the eyes, ears and skin [23,24]. In addition, studies in the USA and China have found an increased risk of low-birth-weight outcomes and live births with birth defects in women in coal mining regions [25,26,27]. Although most of these studies followed an ecological design, the higher morbidity in coal mining areas in countries in North America and Europe and China suggests that populations in regions with large coal mining activities, such as Queensland in Australia, are at risk of higher incidence of various health conditions associated with exposures to coal mining releases.

Australia is the fourth biggest coal producer globally with Queensland, the main coal mining state, having a total output of 1.2 billion tons of coal mined between 2015 and 2020 [28,29]. In this period, 53 mines have been operating in the state, including two of the largest coal mines worldwide [29,30]. Despite the volume of coal production historically in Queensland, there is little research on the association of coal mining emissions and releases with health outcomes in populations in the vicinity of the mines. Some studies have identified health problems that range from mental and social health conditions to a deficiency in access to health care [31,32]. However, these studies focused mostly on coal workers and their families rather than wider exposed populations in coal mining areas, and the links between coal mining exposures and other diseases, such as cardiovascular and respiratory diseases, in the general population are not yet known.

This study explores the association of coal mining exposures with morbidity in populations in the Queensland Local Government Areas (LGAs). The analysis considered the higher concentration of the Queensland population around urban areas and the large expansion of coal mines in rural and less populated areas by implementing a Bayesian spatial regression. Spatial regression models incorporate the spatial correlation between geographical areas to provide robust risk estimates and better quantification of their uncertainty; an approach increasingly used in epidemiological studies of mortality and morbidity in Queensland [33,34,35,36]. This study estimated the association of coal production with hospitalisations due to six disease groups in Queensland, using a Bayesian spatial hierarchical analysis.

## 2. Materials and Methods

This study was approved by the University of Queensland Research Ethics Committee (16 November 2016) with ethics approval granted by the Children’s Health Queensland Hospital and Health Service Human Research Ethics Committee: HREC/16/QRCH/320. The study followed an ecological design, using the Queensland LGAs as the ecological units to implement a cross-sectional analysis of LGA-level hospitalisations merged with surface coal mining coal production in million tons (Mt) and sociodemographic as well as environmental risk factors per LGA, for the period 1996–2010.

### 2.1. Data

Hospitalisation data were obtained from the Queensland Hospital Admitted Patient Data Collection (QHAPDC) that collects data on all admitted patients separated from both public and licensed private hospitals and private day surgeries in Queensland. Six disease groups were determined to have a higher incidence in non-occupational populations in the vicinity of coal mining, from an analysis of studies summarising the evidence on morbidity associated with coal mining [3,13,14]. These disease groups include: circulatory, respiratory, chronic respiratory and hypertensive diseases, as well as lung cancer and diabetes mellitus [13,22]. The number of hospitalisations for each disease group were extracted from the data using the International Classification of Diseases (ICD) codes in the QHAPDC. Each hospitalisation was geocoded to an LGA-map defined elsewhere [37]. In brief, LGA boundaries in the census years 1996, 2001 and 2006 were assessed to produce a map with LGA boundaries consistent across the whole study period 1996–2010. Two 2006-LGAs with very small populations and non-overlapping boundaries were collapsed to overlap the boundaries of a larger 2001-LGA. The LGA-map included 125 LGAs with an average area of 13,839 km^2^ (quartile-1: 1443 km^2^, quartile-3: 16,349 km^2^). There are important differences in the age structure of the population composition between the Queensland LGAs, therefore standardised (i.e., age-adjusted) hospitalisation rates (sHR) were calculated to account for the potential confounding effect of these differences, using the indirect standardisation method [38]. The 2001 census data was used for the standard population following the Australian Bureau of Statistics (ABS) recommendation to use this as the reference population for demographic statistics until 2018 [39]. Coal production (i.e., coal gross raw output) in Mt from open surface coal mines was calculated at the LGA-level using data from the Department of Natural Resources and Mines [40] (details in Appendix A).

The ecological design of the study involves a potential risk of ecological bias, therefore sociodemographic and environmental factors were included to adjust for potential confounders. Sociodemographic data taken from the ABS included the count of indigenous people and people employed in the mining industry and the Index of Socioeconomic Disadvantage (ISD). The ISD summarises several measures about the economic and social conditions of the LGAs population, such as household income, people with no qualifications, or people in low skill occupations [39]. A low score indicates greater socioeconomic disadvantage in an LGA, relative to the whole of the State. To account for gender differences, standardised rates of hospitalisation for each sex were included, as there were important differences related to the age structure for each sex [41]. Standardised rates of the indigenous population and mining employees and the sHR for each gender were calculated using the same method of the sHR, as using a similar standardisation for independent variables is a validated approach to reduce the risk of ecological bias [42] (standardised rates of hospitalisation for each sex in the Appendix A). Population density was calculated as the total LGA population per area (in km^2^). The rate of hospitals per population were calculated using hospital data from the National Health Performance Authority [43]. The average temperature per LGA for the study period was calculated using data from the Australian Bureau of Meteorology (BOM) [44]. No multicollinearity of the independent variables was verified with a variance inflation factor (VIF) test where collinearity was considered for a VIF ≥ 8 [45].

### 2.2. Analysis

A Bayesian hierarchical regression model was used to estimate the hospitalisation risk for each disease group, considering the spatial distribution of the Queensland LGAs. Bayesian spatial models can reduce the estimated variance between geographical areas with small populations and make it easier to assess the prediction of uncertainty based on maximum likelihood and are a robust approach for spatial analyses of Queensland geographical areas [46,47]. The model was fit with the R-INLA package that uses the Integrated Nested Laplace Approximation (INLA) [48]. This is an efficient alternative to computationally and time intensive simulations using Markov Chain Monte Carlo methods to produce robust regression estimates in analyses of spatially auto correlated data [49].

For the i-th LGA, the age-adjusted count of hospitalisations was modelled as
$$y_{i}\sim Poisson\left(\lambda_{i}\right)$$
where the mean $\lambda_{i}$ is defined in terms of a rate $\rho_{i}$ for the population of each LGA. To map the risk of hospitalisation for each LGA, the linear predictor was defined on the logarithmic scale
$$\eta_{i}=\log\left(\rho_{i}\right)=\alpha +s_{i}+u_{i}\;\left(\mathrm{Disease}\;\mathrm{mapping}\;\mathrm{model}\right)$$
where α is the intercept and the parameters s and u represent the spatial structure and the unstructured component (i.e., random effects) according to the Besag-York-Mollie (BYM) specification [47]. The spatial structure was defined as an adjacency matrix with a queen specification (i.e., all neighbours with sharing boundaries), an optimal specification tested in spatial analyses of Queensland LGAs [37].

To estimate the association of the sHR for each disease group with coal production, adjusting for sociodemographic and environmental covariates, the linear predictor was defined as
$$\eta_{i}=\log\left(\rho_{i}\right)=\alpha +\beta_{cp}CP_{i}+\beta_{x}X_{xi}+s_{i}+u_{i}\;\left(\mathrm{Ecological}\;\mathrm{model}\right)$$
where CP is the surface coal mining coal production in Mt and X represents the vector of sociodemographic and environmental covariates with their respective regression coefficients $\beta_{x}$. All predictors were scaled for computational efficiency and ease of interpretation.

Bayesian analyses incorporate priors for estimating parameters and for drawing statistical conclusions. A prior distribution assigns a probability to the value of a parameter to be estimated [50]. As the BYM model incorporates priors on the log of the structured and unstructured effect precisions (s and u), a sensitivity analysis was done to compare the effect of the prior on the regression estimates. Two non-informative priors previously assessed in analyses of Queensland geographical areas [46] and the default R-INLA non-informative prior [50] were compared. This analysis identified the prior that produced the best-fit model for each disease group using the deviance information criterion (DIC) for comparison (details in Appendix A). Important associations were highlighted when the regression coefficient’s credible intervals (95% CI) did not cross the null value of 1. The LGA-specific posterior means were used to map the LGA-specific risk of hospitalisation (residual relative risk of hospitalisation for each LGA compared to the whole of Queensland). Finally, the LGA-specific risk of hospitalisation estimated in the ecological model was assessed only in the LGAs with coal mining activities. The maps were drawn with the R-package T-map [51].

## 3. Results

The study cohort consisted of 2,705,245 hospitalisations across the 15-year (1996–2010) study period (1,206,635 hospitalisations in females and 1,498,610 in males). Table 1 shows the summary statistics of the standardised hospitalisation rate for each disease group.

There were 16 LGAs with coal mining activities in both central east and southeast Queensland (Figure 1), with an average coal production of 19.77 Mt in the study period. The summary statistics for coal production and the sociodemographic and environmental covariates are shown in Table 2.

Table 3 shows the exponentiated posterior mean of the spatial regression estimates in the ecological model for each disease group. A positive association of coal production with the sHR was found for circulatory diseases (1.022, 99% CI: 1.002–1.043) and respiratory diseases (1.031, 95% CI: 1.001–1.062). On the original scale, the regression coefficients mean that each Mt of coal mined was associated with an increase of 81.7 and 82.4 hospitalisations per 1000 people in the study period, for circulatory and respiratory diseases respectively.

The spatial distribution of the LGA-specific risk of hospitalisation is shown in Figure 1. A higher risk of hospitalisation in LGAs grouped in northern, southern and central Queensland was more evident for diabetes mellitus and circulatory diseases and cancer of the bronchus and lung respectively. There were important differences in the spatial distribution of the risk of hospitalisation for all disease groups between the disease mapping model compared with the ecological model. Once the association of coal production and sociodemographic factors were taken into account (i.e., ecological model), LGAs with a higher risk of hospitalisations were found in northwest and central Queensland for circulatory, respiratory and chronic lower respiratory diseases, and in north and southeast Queensland for cancer of the bronchus and lung, respectively. A higher risk of hospitalisation for hypertensive diseases and diabetes mellitus was identified in LGAs in North and West Queensland, respectively.

The LGA-specific risk of hospitalisation estimated in the ecological model, only for the LGAs with coal mining activities is shown in Figure 1 under the coal mining LGAs maps. Most coal mining LGAs are found in central and southeast Queensland. The coal mining LGAs with an increased risk of hospitalisation were: 7 (44%) for circulatory diseases, 9 (56%) for respiratory diseases, 11 (69%) for chronic lower respiratory diseases, 9 (56%) for hypertensive diseases, 8 (50%) for lung cancer and 8 (50%) for diabetes mellitus. Duaringa and Emerald had an increased risk of hospitalisations for all disease groups while an increased risk of hospitalisations for all disease groups except one was found in Banana (except diabetes mellitus), Bauhinia (except diabetes mellitus), and Ipswich (except hypertensive diseases). All other coal mining LGAs had an increased risk of hospitalisation for four or less disease groups. Four coal mining LGAs had an increased risk of hospitalisation for only one disease group: Wambo (lung cancer), Rosalie (chronic lower respiratory diseases), Peak Downs (respiratory diseases), and Belyando (hypertensive diseases), and Nebo had no risk of hospitalisations for any of the disease groups (the LGA-specific risk of hospitalisation in the ecological model for each disease group is shown in the Appendix A).

## 4. Discussion

This study found a positive though small association of coal production with the standardised rate of hospitalisation due to circulatory and respiratory diseases in Queensland. The spatial Bayesian analysis allowed the identification of multiple LGAs with a higher risk of hospitalisation due to these diseases in north and southeast Queensland. We also identified specific LGAs with a higher risk of cancer of the bronchus and lung, hypertensive diseases and diabetes mellitus, grouped in north and southeast, north, and west regions of the state, respectively. However, we did not find evidence of an association between coal production and hypertensive and chronic lower respiratory diseases, diabetes mellitus or cancer of the bronchus and lung for the whole of Queensland. Duaringa and Emerald were identified as the LGAs with coal mining activities (coal mining LGAs) that presented a higher risk of hospitalisations for all disease groups and Nebo was the only coal mining LGA without an increased risk of hospitalisation for any disease group—although these associations in Nebo were not strong (credible intervals < 95%).

The higher risk of circulatory and respiratory hospitalisations associated with coal mining in Queensland found in this study concurs with previous research that has identified an increased risk of morbidity in coal mining areas in America and Europe. Coal production has been associated with a higher risk of hypertension and chronic obstructive pulmonary disease (COPD) in people in coal mining counties in the USA [22] and increased medical consultations due to respiratory symptoms and asthma were estimated in children living near open cast mines in the United Kingdom [24]. Although our analysis found an increased risk of hospitalisation for circulatory and respiratory diseases in the whole of Queensland, the differences between LGAs could be identified in the random effects estimates (i.e., LGA-specific risk). The identification of the health risk associated with coal mining for specific geographical areas has been done in some studies in the USA [17,19,52] but very few studies have investigated these associations in non-occupational exposed populations in Australia, oraddressed larger areas or single communities rather than all of Queensland’s geographical districts [53,54,55,56]. We found important differences between all six disease groups across the coal mining LGAs, with two of these (Duaringa and Emerald) having an increased risk of hospitalisation for all disease groups, although one coal mining LGA with one of the highest levels of coal production (Nebo) had no increased risk of hospitalisation for any of the diseases studied.

These differences can be related to the distribution of sociodemographic risk factors with important determinants of morbidity associated with coal mining, such as socioeconomic status being less defined in the Queensland LGAs compared to district areas in other coal mining regions, such as the USA [57]. There are very small differences in the socioeconomic disadvantage between all LGAs and especially the coal mining LGAs (with less than 10% difference between quartiles 1 and 4 of the ISD). The size and population density of the Queensland LGAs also have significant variations, with a ratio smaller/larger LGA = 1.2 × 10^−4^ and most of the population in LGAs near urban centres and or close to coastal regions. An important characteristic of the Bayesian spatial models used in this study is the identification of the risk variability between LGAs. This can support the design and implementation of public health strategies to protect exposed populations and to improve the characterisation of health impacts required for the coal mining industry sector [58,59]. However, analyses in smaller geographical areas would be required to increase the spatial resolution of the estimates to identify specific communities at risk.

The higher risk of morbidity associated with coal mining in Queensland, particularly for circulatory and respiratory diseases, can be associated with the higher levels of air pollutants in Australian coal mining areas, especially particle matter [6,60]. Fine particle matter (PM_2.5_) can deposit in the smallest sections of the respiratory system causing alveolar damage and can access the blood circulation to be uptaken by the vascular endothelium which can lead to endothelial dysfunction, both of which are mechanisms associated with respiratory and cardiovascular disease [5]. High levels of PM_2.5_ are also linked to an altered inflammatory response associated with higher blood levels of cytokines, such as interleukin-6, interleukin-8 and tumour necrosis factor alpha that induce cellular necrosis or apoptosis and affect the transcription of genes [11,12]. Queensland has the second highest average ambient levels of particulate matter associated with coal mining in Australia [61] which can be linked to the higher risk of circulatory and respiratory diseases identified in our analysis.

Other airborne contaminants including toxic metals can be associated with these diseases in populations in coal mining areas. According to the National Pollutant Inventory, Queensland has the highest loads of selenium (Se), arsenic (As) and volatile organic compounds (VOCs) from coal mining in Australia and the second highest loads of nitrogen dioxides, lead (Pb) and sulphur dioxide (SO_2_) [61]. Exposure to As through contaminated water is associated with lung dysfunction and respiratory disorders in children and adults [62,63,64], and higher blood and cellular levels of VOC metabolites and Se have been associated with endothelial dysfunction, a key determinant of circulatory diseases, in epidemiological and in-vitro studies [65,66]. Multiple studies have demonstrated the association of Pb exposures with respiratory symptoms and diseases, such as asthma and respiratory infections [67,68,69], and increased levels of clinical and serological markers of circulatory diseases [70,71]. The higher coal production levels associated with the increased risk of respiratory and circulatory diseases found in this study can be associated with higher levels of air pollutants, including PM_2.5_ and SO_2_, toxic metals and VOCs that play an important role in the development of these diseases. The proximity to coal mining activities could be associated with other overlapping paths of exposure, such as dust from blasting and traffic and transportation. This can help explain the potential links of coal mining releases with these diseases in Queensland, and their spatial trend with an increased risk of hospitalisation in several coal mining LGAs in central Queensland.

Whereas we found multiple coal mining LGAs with a higher risk of circulatory and respiratory diseases, the specific-LGAs analysis also identified a higher risk of chronic lower respiratory diseases, diabetes mellitus and cancer of the bronchus and lung in coal mining LGAs in central and southeast Queensland. The association of coal mining with lung cancer has been identified in multiple studies of occupational and non-occupational exposed populations [3,13,72]. Exposure to particulate matter released in open-pit coal mines has been associated with chromosomal damage and genetic instability which are determinants of a higher risk of cancer in communities in coal mining areas [73]. Risk assessment studies have also estimated an increased risk of cancer in general populations exposed to coal mining, mediated by PM_10_-and Pm_2.5_-bound trace metals and polycyclic aromatic hydrocarbons, via ingestion, inhalation and or dermal absorption [9,74]. In addition, chronic respiratory diseases, such as COPD, have been associated with chronic exposure to coal mining in other coal mining regions [22] and long-term exposure to NO_2_, one of the biggest coal mining emissions in Queensland [61,75]. Other metals emissions with a high intensity associated with coal mining in Queensland, especially As, are associated with an increased risk of diabetes mellitus and metabolic syndrome [76] which concurs with previous research that identified a higher risk of metabolic diseases in other coal mining regions [21]. The above studies support the possibility that the increased risk of hospitalisation for multiple diseases groups in Queensland found in this study could be associated with releases from coal mining activities.

### Limitations

The limitations of the data determined our design of an ecological study where hospitalisation counts were aggregated at the LGA level. This implies a risk of ecological bias (i.e., analysis of data at the group level can produce spurious associations). To reduce this risk of bias, in addition to adjusting for sociodemographic and environmental factors, the covariates expressed as rates used the same standardisation as the dependent variable, and the Bayesian spatial regressions incorporated a mixed effects model, both of which are approaches that can reduce the risk of ecological bias [77,78]. However, although this analysis estimated a statistical association of coal production with circulatory and respiratory diseases in Queensland, our findings do not provide a measure of causality between coal mining exposures and these diseases. Further research with data at the individual level is required to identify the causal relationship between coal mining and these diseases in Queensland.

Whereas the Bayesian spatial approach could identify the spatial distribution of morbidity risk for all disease groups, it did not address statistical associations potentially changing over time as we implemented an aggregated time series analysis considering the small number of hospitalisations per year for some of the disease groups (e.g., hypertensive and chronic lower respiratory diseases, cancer of the bronchus and lung and diabetes mellitus). This allowed having a larger number of hospitalisations in each geographical area to increase the robustness of the estimates, but we could not identify temporal trends of hospitalisations. In addition, we did not include periodic changes in the production of coal because data on coal production across the study period were limited and have some inconsistencies if measured for different time periods. Further research including temporal changes in hospitalisations and coal production is required to identify the potential effect of periodic variations in coal production on hospitalisations in Queensland.

## 5. Conclusions

A higher production of coal from surface coal mining was associated with a small increase in the risk of hospitalisation for circulatory and respiratory diseases in Queensland, after taking sociodemographic and environmental factors into account. Local Government Areas with coal mining activities in central Queensland were identified to have a higher risk of hospitalisation for these diseases as well as chronic lower respiratory diseases, diabetes mellitus and cancer of the bronchus and lung. A Bayesian spatial regression analysis was used to estimate the risk in specific geographical areas which can be an important tool to support public health strategies to protect at-risk populations and strengthening the assessment of health impacts of coal mining. Further research on individual responses to coal mining exposures and higher spatial and temporal resolution level is necessary to investigate the causal links between coal mining and morbidity in non-occupational exposed populations in Queensland.

## Figures and Tables

**Figure 1 ijerph-19-01206-f001:**
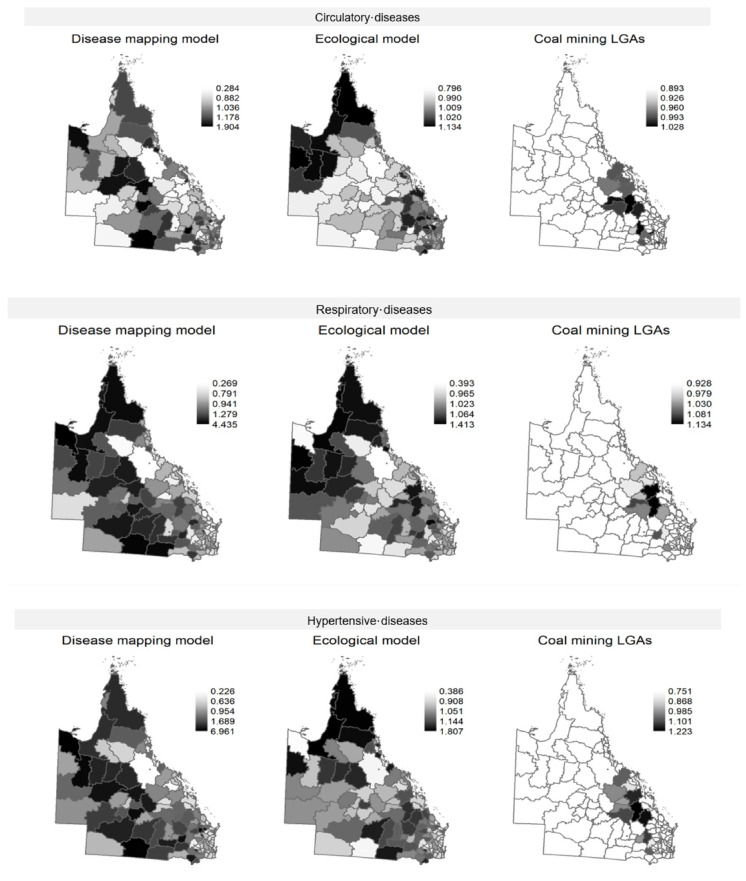
LGA-specific risk of hospitalisation (relative to the whole of Queensland) in the disease mapping and ecological models, and relative risk of hospitalisation estimated in the ecological model for the LGAs with coal mining activities only (coal mining LGAs).

**Table 1 ijerph-19-01206-t001:** Summary statistics of the hospitalisations for each disease group.

Disease Group	ICD-10 Code	Mean	SD	Min	Max
Diseases circulatory system	I00–199	10,030	29,344	52	277,180
Diseases respiratory system	J00–J99	7499	20,702	79	199,305
Hypertensive diseases	I10–I16	208	499	0	4793
Chronic lower respiratory diseases	J40–J47	2267	6449	17	63,423
Diabetes mellitus	E08–E13	1201	3280	3	31,082
Cancer of bronchus and lung	C34	438	1283	0	12,192

SD: Standard deviation. ICD-10: International Classification of Diseases 10th version.

**Table 2 ijerph-19-01206-t002:** Descriptive summary statistics of the predictors.

Predictor	Mean	SD	Min	Max
Coal production (Mt)	19.8	79.9	0	694
Index Socioeconomic Disadvantage	957.7	69.6	472.1	1048.9
Average temperature (°C)	22.2	2.0	16.4	26.2
Population density *	53.7	170.0	0.004	1313.1
Rate of hospitals per population	0.5	0.9	0	6.4
Standardised rate of indigenous population **	78.2	133.4	0	743.5
Standardised rate of mining employees **	23.2	49.2	0	248

* LGA population per area (in Km^2^). ** per 1000 people. SD: Standard deviation.

**Table 3 ijerph-19-01206-t003:** Regression coefficients in the ecological model for each disease group.

**Diseases Circulatory System**			**Diseases Respiratory System**		
	**Posterior Mean * (95% CI)**	**SD**		**Posterior Mean (95% CI)**	**SD**
Intercept	0.354 (0.349–0.359)	1.005	Intercept	0.324 (0.319–0.329)	1.008
Coal production	1.022 (1.002–1.043)	1.008	Coal production	1.031 (1.001–1.062)	1.015
Index of Social Disadvantage	0.999 (0.969–1.03)	1.012	Index of Social Disadvantage	0.99 (0.944–1.039)	1.025
Standardised rate of indigenous population	0.982 (0.949–1.017)	1.014	Standardised rate of indigenous population	0.942 (0.887–0.999)	1.031
Population density	0.998 (0.982–1.013)	1.006	Population density	1.003 (0.977–1.031)	1.014
Rate of hospitals per population	1.01 (0.986–1.034)	1.009	Rate of hospitals per population	1.022 (0.993–1.051)	1.015
Average temperature	1.006 (0.971–1.043)	1.014	Average temperature	0.988 (0.924–1.058)	1.035
Standardised rate of hospitalisation-females	1.132 (1.097–1.168)	1.012	Standardised rate of hospitalisation-females	1.161 (1.07–1.258)	1.042
Standardised rate of hospitalisation-males	1.183 (1.15–1.217)	1.011	Standardised rate of hospitalisation-males	1.344 (1.247–1.447)	1.039
Standardised rate of mining employees	0.965 (0.942–0.988)	1.009	Standardised rate of mining employees	0.968 (0.935–1.002)	1.018
**Hypertensive diseases**			**Chronic lower respiratory diseases**		
	**Posterior mean (95% CI)**	**SD**		**Posterior mean (95% CI)**	**SD**
Intercept	0.012 (0.011–0.012)	1.021	Intercept	0.099 (0.097–0.1)	1.009
Coal production	1.039 (0.977–1.106)	1.032	Coal production	1.025 (0.996–1.054)	1.014
Index of Social Disadvantage	0.982 (0.887–1.087)	1.053	Index of Social Disadvantage	0.965 (0.923–1.009)	1.023
Standardised rate of indigenous population	0.919 (0.818–1.033)	1.061	Standardised rate of indigenous population	0.944 (0.896–0.994)	1.026
Population density	0.995 (0.946–1.047)	1.026	Population density	1.007 (0.985–1.03)	1.011
Rate of hospitals per population	1.058 (0.977–1.142)	1.041	Rate of hospitals per population	1.021 (0.988–1.054)	1.017
Average temperature	0.978 (0.86–1.114)	1.068	Average temperature	0.989 (0.937–1.044)	1.028
Standardised rate of hospitalisation-females	1.559 (1.397–1.741)	1.058	Standardised rate of hospitalisation-females	1.18 (1.134–1.228)	1.020
Standardised rate of hospitalisation-males	1.263 (1.133–1.409)	1.057	Standardised rate of hospitalisation-males	1.34 (1.285–1.397)	1.021
Standardised rate of mining employees	0.969 (0.898–1.044)	1.039	Standardised rate of mining employees	0.971 (0.939–1.003)	1.017
**Diabetes mellitus**			**Cancer of bronchus and lung**		
	**Posterior mean (95% CI)**	**SD**		**Posterior mean (95% CI)**	**SD**
Intercept	0.047 (0.046–0.048)	1.011	Intercept	0.015 (0.014–0.015)	1.012
Coal production	1.032 (0.999–1.065)	1.016	Coal production	1.001 (0.977–1.026)	1.013
Index of Social Disadvantage	0.966 (0.922–1.011)	1.024	Index of Social Disadvantage	0.987 (0.959–1.016)	1.015
Standardised rate of indigenous population	0.881 (0.823–0.943)	1.035	Standardised rate of indigenous population	0.977 (0.937–1.017)	1.021
Population density	1.002 (0.979–1.025)	1.012	Population density	0.992 (0.985–0.999)	1.004
Rate of hospitals per population	1.005 (0.96–1.05)	1.023	Rate of hospitals per population	0.969 (0.922–1.016)	1.025
Average temperature	1.013 (0.958–1.071)	1.028	Average temperature	1.003 (0.981–1.026)	1.011
Standardised rate of hospitalisation -females	1.276 (1.214–1.341)	1.026	Standardised rate of hospitalisation -females	1.232 (1.194–1.27)	1.016
Standardised rate of hospitalisation -males	1.323 (1.269–1.38)	1.022	Standardised rate of hospitalisation -males	1.327 (1.3–1.354)	1.010
Standardised rate of mining employees	0.949 (0.913–0.986)	1.020	Standardised rate of mining employees	0.981 (0.951–1.012)	1.016

Notes. * denotes coefficients have been exponentiated, CI: Credible Interval. SD: Standard deviation.

## Data Availability

Some of the data presented in this study are available on request from the corresponding author conditioned on the restrictions of the ethics approvals. The hospitalisation data are not publicly available due to ethical considerations in the National Statement on Ethical Conduct in Human Research.

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
