# Peer review of "Mapping the Morbidity Risk Associated with Coal Mining in Queensland, Australia"

_ijerph, 2022, doi:10.3390/ijerph19031206_

Round 1
Reviewer 1 Report
Mapping the morbidity risk associated with coal mining in 2 Queensland, Australia
General Comments
This study used Bayesian spatial modes to examine the association of coal production with hospitalizations due to six outcomes (Diseases of circulatory system; respiratory system ; hypertension; Chronic lower respiratory disease; diabetes; and lung cancers) in Queensland. Positive associations were observed between hospitalization rates due to diseases if circulatory and respiratory system. Findings inform public health practices on the association of coal mining practices and adverse health outcomes.
Specific comments
Please include ICD-10 codes for disease groups included in Table 1.
Are numbers reported in Table 1 rates per 100,000 people? Rates related to diseases of the circulatory and respiratory system seems too high (for example 277,180 per 100,000). The authors need to double check these numbers.
Tables need to be stand alone. Please report unit of measures for all variables (i.e., temperature, population density).
Additional information should be provided in the methods section of index of socioeconomic disadvantage; how it is calculated; what does a value of 957.7 mean.
There is no mention of standardized rate of hospitalization for males vs females until the results of Bayesian models are reported. No descriptive statistics on this variable is reported. I am not sure why models should include standardized rate of hospitalization for both males and females. First, the outcomes and this independent variables are correlated, secondly if the aim of the study is to examine difference in rates of disease between males or females, separate models needs to be generated for different sexes..
Table 3. mean and 95%CI for the mean is provided. I am not sure SD needs to be included. Authors are more than welcome to keep or remove SDs.
The study period is 1996-2010. Have coal mining practices changed (Expanded/shrank) during this time frame? This point should be included as a limitation in the discussion section because coal mining practices and locations may be longitudinally different and its effect on population may chance over time. There is some discussion of needing a time-series analysis in the limitation section but it needs to be expanded and include change in mining practices as well.
In addition, migration might have happened during this study. People who lived closer to coal mining activities might have relocated. This is another limitation of this study.
Author Response
Reviewer 1.
General Comments
This study used Bayesian spatial modes to examine the association of coal production with hospitalizations due to six outcomes (Diseases of circulatory system; respiratory system; hypertension; Chronic lower respiratory disease; diabetes; and lung cancers) in Queensland. Positive associations were observed between hospitalization rates due to diseases if circulatory and respiratory system. Findings inform public health practices on the association of coal mining practices and adverse health outcomes.
Specific comments
- Please include ICD-10 codes for disease groups included in Table 1.
R. We agree the ICD codes are necessary and need to be included. We have added the ICD-10 codes for each disease group in table 1
___________
- Are numbers reported in Table 1 rates per 100,000 people? Rates related to diseases of the circulatory and respiratory system seems too high (for example 277,180 per 100,000). The authors need to double check these numbers.
R. Thanks for noting this. There was a mistake in the title. The table shows the summary statistics of hospitalisations for each of the disease groups. We have verified these figures and corrected the title in Table 1.
___________
- Tables need to be stand alone. Please report unit of measures for all variables (i.e., temperature, population density).
R. Thanks for this comment. We have updated Table 2 with the unit measures of the average temperature and population density
___________
- Additional information should be provided in the methods section of index of socioeconomic disadvantage; how it is calculated; what does a value of 957.7 mean.
R. The Index of Socio-economic Disadvantage (ISD) summarises several measures of the Australian Bureau of Statistics -ABS about the economic and social conditions of the LGAs population such as household income, people with no qualifications, or people in low skill occupations [1]. A low score indicates greater socioeconomic disadvantage in a LGA relative to the whole of the State. We have included an additional explanation of this index and its interpretation in the manuscript with reference to the technical aspects considered by the ABS (changes in lines 121-124)
___________
- There is no mention of standardized rate of hospitalization for males vs females until the results of Bayesian models are reported. No descriptive statistics on this variable is reported. I am not sure why models should include standardized rate of hospitalization for both males and females. First, the outcomes and this independent variables are correlated, secondly if the aim of the study is to examine difference in rates of disease between males or females, separate models needs to be generated for different sexes.
R. We did not include these rates in the first version of the manuscript for practical reasons (the table includes summary statistics for each of the diseases groups separately for females and males). We appreciate your comment and have reconsidered this. The descriptive statistics of the standardized rate of hospitalization (for each disease group and sex) has been included in the supplementary material with a specific mention of it in the he revised manuscript (Lines 129-130).
We included these rates in the model to adjust for differences in the hospitalisations for each sex since both rates can be credible predictors [2]. There were important differences related to the age structure for each sex (the standardisation method included a different standard population for each sex: population of females and males per LGA, respectively -the standardisation method is further explained in lines 126-129 of the revised manuscript). While the inclusion of the male/female hospitalisation proportion in regression models is a common practice, we opted to follow the approach proposed by Rosenbaum and Rubin [3] considering the potential risk of ecological bias. In this case, the standardisation method used for the covariates should be the same as used for the dependent variable, an approach that has been reiterated in further analyses to assess and reduce the risk of ecological bias [4]. In addition, we considered the potential correlation for these covariates and run a Variance Inflation Factor test to verify a condition of no collinearity [5]. An explanation of these previous analyses and the inclusion of standardized rate of hospitalization for males and females as covariates have been incorporated in the revised manuscript in Lines 124-26; 134-136.
___________
- Table 3. mean and 95%CI for the mean is provided. I am not sure SD needs to be included. Authors are more than welcome to keep or remove SDs.
R. All co-authors have agreed to include the standard deviation in all tables in the manuscript.
___________
- The study period is 1996-2010. Have coal mining practices changed (Expanded/shrank) during this time frame? This point should be included as a limitation in the discussion section because coal mining practices and locations may be longitudinally different and its effect on population may chance over time. There is some discussion of needing a time-series analysis in the limitation section but it needs to be expanded and include change in mining practices as well. In addition, migration might have happened during this study. People who lived closer to coal mining activities might have relocated. This is another limitation of this study.
R. Thanks for your comment. We agree that changes in the production of coal between years can have an impact on the effect of this variable in the analysis. We did not include a temporal dimension in the regression analysis mostly because yearly data on coal production for the mining leases across the study period were limited and/or with some inconsistencies. Given this limitation and the small number of hospitalisations per year for some of the disease groups, we decided to do an aggregated time series analysis rather than incorporating a temporal component. This, in turn, affected the chance to explore the effect of demographic changes in the population, such as migration trends between years. We agree that these limitations need to be addressed and we have included them in a new section in the discussion of limitations (in lines 366-371).
___________
References
- ABS, 3101.0 - Australian Demographic Statistics, Sep 2018. Retrieved from https://www.abs.gov.au/Ausstats/abs@.nsf/glossary/3101.0, 2018.
- Kronmal, R.A., Spurious correlation and the fallacy of the ratio standard revisited. Journal of the Royal Statistical Society: Series A (Statistics in Society), 1993. 156(3): p. 379-392.
- Rosenbaum, P.R. and D.B. Rubin, Difficulties with Regression Analyses of Age-Adjusted Rates. Biometrics, 1984. 40(2): p. 437-443.
- Milyo, J. and J.M. Mellor, On the Importance of Age-Adjustment Methods in Ecological Studies of Social Determinants of Mortality. Health Services Research, 2003. 38(6p2): p. 1781-1790.
- O’brien, R.M., A caution regarding rules of thumb for variance inflation factors. Quality & quantity, 2007. 41(5): p. 673-690.
Reviewer 2 Report
- Hospitalization data only come from inpatients in Queensland Hospital, so why can it represent all hospitalizations in Queensland?
- There are many diseases that may be associated with it, and there are also many other diseases with high morbidity. Why choose these six diseases?
- According to lines 102-103, you mentioned age adjustment. What are the criteria for age adjustment and on what basis? Age is also one of the factors related to the incidence of many diseases. There is little explanation for the age attribute of the sample data. Should this attribute be added?
- Why choose the A Bayesian hierarchical regression model model? Have the results of this model been tested?
- The incidence of respiratory diseases varies from season to season. Your research doesn't seem to take into account the scale of time. It is suggested whether it is possible to add the scale of time and space while studying the correlation.
- The information in the table, such as Average temperature, has no units.
- The DOI numbers of some references are not shown in the manuscript. Please add it.
Reviewer 3 Report
Intro- Line 36: It is appropriate to mention the occupational health impacts of coal mining, which you do. "...especially in coal miners who have an increased risk of chronic respiratory and lung cancer and occupational diseases such as silicosis." The older characterization was coal miners pneumocysticosis. The growing recognition that both coal dust and silica contribute has led to the adoption of "coal mining dust lung disease" (CMDLD) as the accepted term, which includes a spectrum of diseases (Petsonk, 2013). It would be good to add a sentence or so to briefly describe the effects of occupational exposure.
Line 56: It should be noted that the studies associating coal production with birth defects are ecological. Same with some of the others, which have a lower level of evidence for causation.
Methods: Are solid and modelling appears well thought out.
Line 151: This paragraph would use a more detailed explanation of the analysis. Could you expand on what "priors" are?
Results-
Line 182: Are you sure the rate was hospitalizations per 1000 people? I would have expected per 10,000 people.
Line 200: This sentence would be more clear if you used either a comma or parenthesis for the phrase "mostly located in central Queensland,".
Discussion- Line 227: For people not familiar with Australia, it might be helpful to add in this sentence that southeast Queensland is coal producing.
Line 242: Should "consultants" be "consultations"?
Line 255: This is a very interesting and important point. Most ecologic studies in the US have been in the Appalachian coalfields which are marked by high rates of poverty and smoking, which makes interpretation of ecologic studies more challenging. That this study was done in a region with more narrow distribution in socioeconomic factors suggests the findings are less likely to be due to confounding.
Paragraphs 269 & 282: Very good, succinct discussion of PM and metals exposure and effects. In considering putative exposures it might be worthwhile to mention that proximity to surface coal mining carries multiple overlapping routes of exposure for communities (dust from blasting, truck and rail traffic, dust from coal preparation plants etc.). I mention this from the perspective of a community member.
Line 309: Do you mean dermal absorption?
References-
Line 405: Kentucky and Appalachia are capitalized.
Reviewer 4 Report
Estimated Authors, I've read with great interest the present paper entitled "Mapping the morbidity risk associated with coal mining in Queensland, Australia". Here you'll find my report.
Outline of the paper. Cortes-Ramirez et al. have summarized available data on a total of 2,705,245 hospitalizations across 15 years (i.e. 1996 to 2010). An analysis performed by means of a Bayesian hierarchical regression model estimated the hospitalization risk for several disease groups that were assumed to be related with the non-occupational exposure to coal mining in a state of Australia (i.e. Queensland) were residential exposure may be acknowledged as particularly significant. Briefly, the study found a positive but small association of coal production with hospitalization rates due to circulary and respiratory diseases.
Such results are consistent with current understanding of the impact of coal mining pollution on the respiratory system, as properly discussed by study Authors.
Analysis of the study:
- introduction: it is informative, as it provide a concise but clear description of the background of this study; aims are clearly described;
- Materials and Methods: authors provide appropriate information on the source of their data, ethical approval, and statistical analysis. On this regard, Authors have also provided an accurate discussion on the limits of their Bayesian approach to data analysis.
- Results: concise but informative and clearly provided.
- Discussion: consistent with the results, such section also compares the results with available evidence, not only from epidemiological point of view, but enforcing the estimates through the discussion of alleged mechanisms of diseases.
In summary: a very well written paper, reporting on a topic of certain interest. I endorse its acceptance as it is.
Author Response
Comments and Suggestions for Authors
Estimated Authors, I've read with great interest the present paper entitled "Mapping the morbidity risk associated with coal mining in Queensland, Australia". Here you'll find my report.
Outline of the paper. Cortes-Ramirez et al. have summarized available data on a total of 2,705,245 hospitalizations across 15 years (i.e. 1996 to 2010). An analysis performed by means of a Bayesian hierarchical regression model estimated the hospitalization risk for several disease groups that were assumed to be related with the non-occupational exposure to coal mining in a state of Australia (i.e. Queensland) were residential exposure may be acknowledged as particularly significant. Briefly, the study found a positive but small association of coal production with hospitalization rates due to circulary and respiratory diseases.
Such results are consistent with current understanding of the impact of coal mining pollution on the respiratory system, as properly discussed by study Authors.
Analysis of the study:
- introduction: it is informative, as it provide a concise but clear description of the background of this study; aims are clearly described;
- Materials and Methods: authors provide appropriate information on the source of their data, ethical approval, and statistical analysis. On this regard, Authors have also provided an accurate discussion on the limits of their Bayesian approach to data analysis.
- Results: concise but informative and clearly provided.
- Discussion: consistent with the results, such section also compares the results with available evidence, not only from epidemiological point of view, but enforcing the estimates through the discussion of alleged mechanisms of diseases.
In summary: a very well written paper, reporting on a topic of certain interest. I endorse its acceptance as it is.
R. Thank you for your review and interest in our study and manuscript.